# Genetic Dissection of Drought Tolerance of Elite Bread Wheat (*Triticum aestivum* L.) Genotypes Using Genome Wide Association Study in Morocco

**DOI:** 10.3390/plants11202705

**Published:** 2022-10-13

**Authors:** Zakaria El Gataa, Karima Samir, Wuletaw Tadesse

**Affiliations:** 1The International Center for Agricultural Research in the Dry Areas (ICARDA), Rabat 10080, Morocco; 2Faculty of Sciences Ben M’sick, University Hassan II of Casablanca, Casablanca 7955, Morocco

**Keywords:** wheat, drought stress, GWAS, MTA

## Abstract

Drought is one of the most important yield-limiting factors in Morocco. Identification and deployment of drought-tolerant wheat varieties are important to cope with the challenge of terminal moisture stress and increase wheat productivity. A panel composed of 200 elite spring bread wheat genotypes was phenotyped for yield and agronomic traits for 2 years (2020 and 2021) in Morocco under rainfed and irrigated environments. The panel was genotyped using 20K SNPs and, after filtration, a total of 15,735 SNP markers were used for a genome-wide association study (GWAS) using a mixed linear model (MLM) to identify marker-trait associations (MTA) and putative genes associated with grain yield and yield-related traits under rainfed and irrigated conditions. Significant differences were observed among the elite genotypes for grain yield and yield-related traits. Grain yield performance ranged from 0.97 to 6.16 t/ha under rainfed conditions at Sidi Al-Aidi station and from 3.31 to 9.38 t/h under irrigated conditions at Sidi Al-Aidi station, while Grain yield at Merchouch station ranged from 2.32 to 6.16 t/h under rainfed condition. A total of 159 MTAs (*p* < 0.001) and 46 genes were discovered, with 67 MTAs recorded under rainfed conditions and 37 MTAs recorded under irrigated conditions at the Sidi Al-Aidi station, while 55 MTAs were recorded under rainfed conditions at Merchouch station. The marker ‘BobWhite_c2988_493’ on chromosome 2B was significantly correlated with grain yield under rainfed conditions. Under irrigated conditions, the marker ‘AX-94653560’ on chromosome 2D was significantly correlated with grain yield at Sidi Al-Aidi station. The maker ‘RAC875_c17918_321’ located on chromosome 4A, associated with grain yield was linked with the gene TraesCS4A02G322700, which encodes for F-box domain-containing protein. The markers and candidate genes discovered in this study should be further validated for their potential use in marker-assisted selection to generate high-yielding wheat genotypes with drought tolerance.

## 1. Introduction

Drought is the most significant abiotic factor impacting wheat yield and production, particularly in West and South Asia, North Africa, and Sub-Saharan Africa [1]. Climate change reduces precipitation, worsens drought stress, and has a significant influence on wheat production. Drought stress affects wheat grain production and quality at all phases of development and reduces wheat grain yield by around 30% [2,3]. Understanding drought-tolerance processes and identifying loci relevant to drought tolerance are essential for developing breeding tactics that promote drought tolerance [4]. In season 2022/2021, Morocco harvested 5.06 million tons of bread wheat on 3.2 million hectares, with an average of 1.5 tons per hectare.

Using molecular markers such as single nucleotide polymorphism (SNP) and diversity array technologies (DArT), marker-assisted selection (MAS) has become an essential strategy in wheat breeding for the development of novel adapted lines to biotic and abiotic stresses. Based on linkage disequilibrium (LD), a genome-wide association study (GWAS) is a strong method for identifying marker-trait associations (MTA) and quantitative trait loci (QTL) associated with the trait of interest [5]. GWAS employed a variety of statistical models, although the Mixed Linear Model (MLM) and the General Linear Model (GLM) were the most often used. Several studies use MLM as an appropriate model because of including kinship (K) among individuals and population structure (Q) [5].

The primary goal of plant genetics and breeding research is to improve agricultural productivity and stability in order to provide food security for the world’s rapidly rising population. Plant breeding methodologies, genetic designs, genomics, statistics, and bioinformatics must all be combined to adjust adaptive, agronomic, and economic qualities to meet this challenge. Gene identification and complex trait dissection are two linked components of this effort. Complex trait dissection emphasizes genetic contribution and modes of action from numerous loci, which contributes to phenotypic variance, whereas gene identification focuses on individual genes. The two common goals of any QTL mapping in plants are: (1) to increase our biological knowledge of the inheritance and genetic architecture of quantitative traits, both within a species and across related species, and (2) to detect markers that can be used as indirect selection tools in breeding programs [6].

In recent years, significant efforts have been made to complement traditional plant breeding methods based on a phenotypic pedigree with molecular genetic techniques for the discovery and application of QTLs for marker-assisted breeding and genomic selection (GS). Linkage mapping (also known as genetic mapping) and association mapping (linkage disequilibrium (LD) mapping) are the most commonly used approaches in dissecting complex traits and identifying genes underlying trait variation in plants, animals, and humans [7].

Recently, research on association mapping in several agricultural plants has been conducted to address the constraints identified in linkage-based QTL mapping. Fast improvement in genotyping methods, such as genome sequencing and high-density single nucleotide polymorphism, allowed for the creation of association mapping, the mapping of complex characteristics, and the discovery of contributing genes [8,9]. In this study, we carried out GWAS on 200 spring bread wheat genotypes that were phenotyped for grain yield and yield-related traits and genotyped with SNP markers to identify markers as well as to identify putative genes associated with grain yield and yield-related traits under drought and irrigated conditions.

## 2. Results

### 2.1. Phenotypic Description

Figure 1 shows scatter plots of grain yield under rainfed and irrigated conditions at Sidi Al-Aidi station of 200 bread wheat genotypes grown during two seasons (2020 and 2021). The statistical analysis of 200 bread wheat genotypes for grain yield and yield-related traits is presented in Table 1. Grain yield ranged from 3.31 to 9.38 t/h with a mean of 5.93 t/h and the heritability value of 0.45 for irrigated conditions at Sidi Al-Aidi station and grain yield for a rainfed experiment at Sidi Al-Aidi station ranged from 0.97 to 6.16 t/h with a mean of 4.19 t/h and the heritability value of 0.32. While grain yield at Merchouch station ranged from 2.32 to 6.85 t/h with a mean of 4.09 and a heritability value of 0.42. Irrigated conditions at Sidi Al-Aidi station recorded the highest value of grain yield followed by rainfed conditions at Merchouch and Sidi Al-Aidi stations, respectively. All traits showed significant results, wherein the irrigated conditions at Sidi Al-Aidi the NTP value had the highest coefficient of variance (CV) with a value of 23.28% followed by biomass and NSS with a value of 22.68 and 20.59%, respectively. While days to maturity (DMA) recorded the lowest value of CV (2.41%). In the rainfed experiment at Sidi Al-Aidi station, the number of seeds per spike (NSS) had the highest CV with a value of 22.85% followed by biomass and grain yield (GY) with a value of 20.39 and 15.04%, respectively. While canopy temperature (CT) recorded the lowest value of CV (3.57%). In the rainfed experiment at Merchouch station, the GY had the highest CV with a value of 18.29% followed by biomass and number of plants in m^2^ (NP/m^2^) with a value of 16.85 and 14.39%, respectively. While DMA recorded the lowest value of CV (2.97%). Traits in irrigated conditions recorded a higher heritability than in rainfed conditions. Under irrigated conditions at Sidi Al-Aidi station, NSS recorded the highest value of heritability with 0.80 followed by biomass and plant height (PLH) with 0.77 and 0.72, respectively. While GY traits showed the lowest value of heritability. Under rainfed conditions at Sidi Al-Aidi station, PLH recorded the highest value of heritability with 0.65 followed by NSS and days to heading (DHE) with 0.63 and 0.62, respectively. While the GY trait showed the lowest value of heritability. Under rainfed conditions at Merchouch station, Biomass recorded the highest value of heritability with 0.70 followed by PLH and DHE with 0.67 and 0.64, respectively. While the GY trait showed the lowest value of heritability. The top 10 elite genotypes for grain yield are indicated in Appendix A. The genotypes G-146 (SUDAN#3/SHUHA-6/4/BOW/PRL//BUC/3/WH576) and G-145 (STAR *3/LOTUS 5//TNMU/MILAN/3/QAMAR-2) were recorded the highest grain yield under the rainfed condition with a value of 6.16 and 5.75t/h, respectively.

### 2.2. Principal Components Analysis (PCA) and Correlation

Principal component bi-plot analyses show the association between grain yield and yield-related traits (Figure 2). Under the rainfed regime at Sidi Al-Aidi station, the two principal components explained 20.5% and 13% of the total variation for grain yield and yield-related traits. Under the irrigated regime at Sidi Al-Aidi station, the two principal components explained 17% and 15.5% of the total variation for grain yield and yield-related traits. Under the rainfed regime at Merchouch station, the two principal components explained 22.3% and 15.8% of the total variation for grain yield and yield-related traits. Figure 3 shows four groups under rainfed and irrigated regimes at both stations. Figure 4 shows the correlation between agronomic traits under rainfed and irrigated regimes at both stations. Under the rainfed regime at Sidi Al-Aidi station, DHE was positively correlated with days to maturity with a value of 0.53 and biomass was positive with PLH and biomass with a value of 0.31. Whereas, canopy temperature showed a significant negative correlation with grain yield and biomass with values of −0.21 and −0.20, respectively. Under the irrigated regime at Sidi Al-Aidi station, days to heading and days to maturity were correlated with a value of 0.65, grain yield is correlated with biomass with a value of 0.49; also, grain yield was positively correlated with plant height with a value of 0.32. Whereas canopy temperature was negatively correlated with NPM and NSS with a value of −0.18 and −0.15, respectively, and also, grain yield was negatively correlated with days to heading with a value of −0.18.

### 2.3. Linkage Disequilibrium and Marker Trait Associations

Figure 5 shows the LD decay of A, B, and D genomes. Genome B had the highest linkage disequilibrium, with an average r^2^ value of 0.18, followed by genomes A and D with r^2^ = 0.17 and r^2^ = 0.15, respectively. Figure 6 shows the density of markers over 21 chromosomes of 200 bread wheat genotypes. For all traits under the rainfed regime at Sidi Al-Aidi station, a total of 67 significant marker-trait associations (MTAs) with *p* < 0.001 were found (Table 2), the A sub-genome has the most MTAs (28), followed by the B and D sub-genomes, which had 26 and 13 MTAs, respectively. For all traits under the irrigated regime at Sidi Al-Aidi station, a total of 37 significant marker-trait associations (MTAs) with *p* < 0.001 were found (Table 3), the A sub-genome has the most MTAs (16), followed by the D and B sub-genomes, which had 12 and 9 MTAs. For all traits under the rainfed conditions at Merchouch station, a total of 55 significant marker-trait associations (MTAs) with *p* < 0.001 were found (Table 3), the B sub-genome has the most MTAs (21), followed by the A and B sub-genomes, which had 20 and 13 MTAs. The GWAS results is the mean of two years (2020/2021).

The results of significant marker-trait associations for grain yield and yield-related traits under rainfed and irrigated conditions at Sidi Al-Aidi and Merchouch stations are shown in Table 2, Table 3 and Table 4, and Figure 7 as well as Appendix A. For rainfed conditions at Sidi Al-Aidi station, seventy-four MTAs were recorded (Table 2), where 11 MTAs were recorded for biomass, these markers are located on chromosomes 2B, 2D, 3A, 3B, 5D, and 6D, on the same trait the markers “BobWhite_c2988_493” and “Ex_c14755_1362” located on chromosomes 2B and 2D, respectively, with a value of −Log10(p) = 4.17. Twelve markers were significantly associated with canopy temperature located on chromosomes 1B, 2B, 3D, 5B, 6A, 6B, 6D, and 7A, where the marker “AX-94463982” recorded the highest −Log10(p) = 7.03 located on chromosome 7A, whereas the marker “BobWhite_rep_c64068_241” recorded the lowest −Log10(p) = 3.09 located on chromosome 2B. Chlorophyll content recorded three significant markers located on chromosomes 3A, 5A, and 1B. Days to heading recorded two significant markers located on chromosomes 1B and 2B. Days to maturity recorded two significant markers located on chromosomes 5B and 7D. Grain yield recorded three significant markers located on chromosomes 4A, 2B, and 2D. Fifteen MTAs were recorded for NPM, which are located on chromosomes 1A, 1D, 3A, 3D, and 4A, where the marker “Ra_c29200_300” recorded the highest −Log10(p) = 3.63 located on chromosome 1A. For the NSS trait, 11 MTAs were recorded located on chromosomes 1A, 1B, 1D, 5B, 6B, and 7D. The marker “Excalibur_c5329_1335” had the highest −Log10(p) = 4.85. The number of tillers per plant (NTP) trait recorded one significant marker located on chromosome 3B. Seven MTAs were recorded for plant height located on chromosomes 1B, 3B, 4A, 5A, 5B, 5D, and 6A. The marker “RAC875_c41731_321” located on chromosome 2A was recorded for canopy temperature and chlorophyll content.

For irrigated conditions at Sidi Al-Aidi station, forty-eight MTAs (Table 3) were recorded, where 5 MTAs were recorded for biomass, these markers are located on chromosomes 5B and 7A. Chlorophyll content recorded seven MTAs located on chromosomes 1D, 2A, 3A, 3B, 5B, and 7A, where the marker “IAAV5819” recorded the highest −Log10(p) = 3.95 located on chromosome 3B. Seven significant markers were associated with canopy temperature, where the markers were located on chromosomes 1D, 2A, 3A, 3B, 5B, and 7A. DHE recorded one significant marker located on chromosome 1D with −Log10(p) = 3.81. DMA recorded 3 MTAs located on chromosomes 2A and 2D. Six significant MTAs were recorded for NPM located on chromosomes 2A, 2D, 5A, 5B, and 6A, where the marker “Excalibur_rep_c69730_391” had the highest −Log10(p) = 4.87 located on chromosome 5B. NSS trait recorded 5 MTAs located on chromosomes 4A and 5D. NTP trait recorded two significant markers located on chromosome 1B. Plant height recorded one MTA located on chromosome 7A with −Log10(p) = 3.07. The marker “AX-94513795” located on chromosome 3A was recorded for chlorophyll content and biomass.

For rainfed conditions at Merchouch station, fifty-five MTAs were recorded (Table 4), where 3 MTAs were recorded for biomass, these markers are located on chromosomes 6A, 6B, and 6D. Chlorophyll content recorded 12 MTAs located on chromosomes 1A, 1B, 2D, 3B, 3D, 6A, 6D, and 7A, where the marker “AX-94419426” recorded the highest −Log10(p) = 4.34 located on chromosome 1B. Eleven significant markers were associated with canopy temperature, where the markers were located on chromosomes 1A, 1B, 2D, 3A, 3B, 3D, 6A, and 7A. DHE recorded 14 significant markers, the marker “IAAV2346” recorded the highest −Log10(p) = 3.63 located on chromosome 5B. DMA recorded 6 MTAs located on chromosomes 2B, 4B, 4D, 5A, and 5B. Six significant MTAs were recorded for NPM located on chromosomes 3D, 5B, and 7D, where the marker “IACX11794” had the highest −Log10(p) = 4.04 located on chromosome 7D. Thousand kernel weight (TKW) trait recorded 2 MTAs located on chromosomes 2B and 5A. The marker “BS00003816_51” located on chromosome 3A was recorded for canopy temperature and chlorophyll content.

Under rainfed conditions at Sidi Al-Aidi station, out of the two highest significant MTAs found for grain yield, BobWhite_c2988_493 and Ex_c14755_1362 located on chromosomes 2B and 2D, respectively, (Figure 8) expressed the highest frequency of favorable alleles for grain yield in the population. The results showed that wheat genotypes carrying the cytosine and guanine bases at the BobWhite_c2988_493 and Ex_c14755_1362 markers combination, respectively, significantly out-yielded the other base combinations. The combination of the cytosine and thymine bases is shown to have low-yield performance and that is linked to the low cumulative number of favorable alleles of the two significant markers. Under irrigated conditions at Sidi Al-Aidi station, two of the highest significant MTAs found for grain yield, AX-94653560 and wsnp_Ex_c6748_11659366 located on chromosomes 2D and 2B, respectively, (Figure 8) expressed the highest frequency of favorable alleles for grain yield in the population. The results showed that wheat genotypes carrying the cytosine and thymine bases at the AX-94653560 and wsnp_Ex_c6748_11659366 markers combination, respectively, significantly out-yielded the other base combinations. The combination of the thymine and cytosine bases has low-yield performance, which is linked to the low-cumulative number of favorable alleles of the two significant markers.

### 2.4. Gene Annotation

Forty-six putative candidate genes were discovered for all significant markers related to the grain yield and yield-related traits (Appendix A), where 34 candidate genes were recorded for rainfed conditions and 12 genes were recorded for irrigated conditions. In rainfed conditions, for biomass, the highest significant marker on chromosome 2D, ‘Ex_c14755_1362’ was linked to the TraesCS2D02G053300 gene, which encodes the Receptor-like serine/threonine-protein kinase protein involved in serine/threonine/tyrosine kinase activity. Seven genes were identified for canopy temperature, where the highest significant marker ‘AX-94463982’ on chromosome 7A was linked to the TraesCS7A02G088800 gene, which encodes Pre-mRNA-splicing factor 38 involved in the mRNA splicing, via spliceosome. In irrigated conditions, the gene TraesCS5B02G032700 was linked to the marker ‘AX-94826800’ located on chromosome 5B which is associated with chlorophyll content, the gene encodes for Kinesin-like protein involved in microtubule binding. The gene TraesCS5B02G300100 was linked to the marker ‘wsnp_Ex_c6748_11659366’ located on chromosome 5B which is associated with biomass the gene encodes for Trigger_N domain-containing protein involved in peptidyl-prolyl cis-trans isomerase activity.

## 3. Discussion

### 3.1. Phenotypic Variability for Grain Yield and Yield-Related Traits

Bread wheat is a major crop in many parts of the world, particularly in Central and West Asia and North Africa (CWANA) as well as in Sub-Saharan Africa (SSA). In the CWANA region, water shortage is becoming a severe challenge for wheat production. The creation of novel drought-tolerant bread wheat genotypes might be a long-term solution for increasing wheat yield in challenged regions. ICARDA’s bread wheat breeding program targets generating genotypes that are abiotic and biotic stress-resistant and have acceptable end-use quality. Grain yield is a complicated property governed by numerous genes, the expression of which is controlled by environmental conditions. Therefore, this study was carried out to evaluate the drought-tolerance capacity of 200 bread wheat genotypes and to discover markers that are strongly related to grain yield and yield-related traits. Phenotype observations showed that wheat in rainfed conditions had lower GY and fewer DHE than wheat in irrigated conditions. The days to heading and maturity exhibited greater heritability than grain yield at all conditions, as previously reported [35,36]. As a result, grain yield heritability is often lower than that of other characteristics [37]. The correlation between days to heading and maturity was strongly positively correlated in both conditions which were consistent with previous research [38]. Biomass was positively correlated with grain yield in both conditions, but the correlation in irrigated conditions was upper than in rainfed conditions, this result is due to the ample irrigation and optimum conditions.

### 3.2. Marker Trait Association

GWAS has been used in several studies on bread wheat to uncover novel MTAs and QTLs linked with distinct traits in different conditions. We found a total of 104 significant markers under rainfed and irrigated conditions associated with agronomic traits across chromosomes in this study, where 44 MTAs found on the A genome, followed by the B and D genomes with 35 and 25, respectively, similar to another study that found the majority of MTAs for grain yield and yield-related traits on the A and B genomes, followed by the D genome [39]. Drought-stressed environments often have more major MTAs than non-stressed environments, which explains why drought tolerance is highly influenced by genotype by environmental factors [38]. Sixty-seven and thirty-seven MTAs at *p* < 0.001 were found associated with grain yield and agronomic traits under rainfed and irrigated conditions, respectively.

The highly significant marker for grain yield in rainfed conditions was identified on chromosome 2B with −Log10(p) = 3.81 at position 338 Mbp. We identified two other significant markers on chromosomes 2D and 4A. Several studies detected many MTAs/QTLs for grain yield in the same chromosomes [5,35]. In this study, we identified five MTAs for biomass in irrigated conditions, which are located on chromosomes 7A and 5B where the highest marker was ‘BS00022169_51’ located on chromosome 7A with −Log10(p) = 4.14 at position 691 Mbp, a previous study found the same marker on chromosome 7A [40]. On chromosome 5B, a highly significant marker associated with a number of seeds per spike under rainfed conditions was discovered with −Log10(p) = 4.85. Several MTAs/QTLs on chromosome 5B have been linked to NSS in dry conditions, according to a previous study [41]. Five MTAs were found under irrigated conditions located on chromosomes 4A and 5D for grain number per spike. Using 192 bread wheat genotypes planted at the same study site in Morocco, Alemu et al. [42] discovered that the marker ‘wsnp_Ex_c20386_29451037’ was linked to grain number per spike on chromosome 4A. Twelve MTAs regulating canopy temperature traits under rainfed conditions were found on chromosomes 1B, 2B, 3D, 5B, 6A, 6B, 6D, and 7A. MTAs for canopy temperature on chromosomes 7A and 2B were previously reported using different wheat panels cultivated in various settings [43]. Utilizing a population of 287 advanced elite lines of spring wheat, Sukumaran et al. [44] discovered the QTL RAC875_rep_c114561_587 associated with canopy temperature on chromosome 6A. The investigation was conducted using 90,000 wheat SNP created by Infinium iSelect SNP assays. The marker ‘BS00037225_51’ was associated with plant height on chromosome 3B with −Log10(p) = 7.1 at position 590 Mbp. Previously, numerous MTAs/QTLs associated with plant height were discovered on the same chromosome that we discovered [5]. The important markers were mapped to the bread wheat genome reference database at Ensemble Plant and UniProt in order to discover the probable genes linked with the many assessed features previously published. The findings indicated 46 genes involved in ATP binding, mRNA binding, protein serine kinase activity, protein heterodimerization activity, defense response, and oxidoreductase activity, among other biosynthetic pathways [45,46,47].

## 4. Materials and Methods

### 4.1. Plant Material and Experimental Conditions

This study employed a population of 200 bread wheat genotypes from the International Center for Agricultural Research in Dry Areas (ICARDA) (Appendix A). The field experiment was conducted at ICARDA’s experimental stations in Sidi El-Aidi, Morocco (33°07′27.6″ N 7°37′43.8″ W, 406 m a.s.l.) and Marchouch, Morocco (33°36′26.8″ N 6°42′43.9″ W) during the 2019–2020 and 2020–2021 cropping seasons. The genotypes were grown in a 3 m^2^ plot following an alpha lattice design with two replications. In Sidi EL-Aidi station the experiments were carried out in two different water regimes, rainfed and irrigated. For Marchouch station, the experiment was carried out under rainfed conditions. During the growth cycle in the irrigated experiment, the plots were watered by drip irrigation with three irrigations each week. The Sidi El-Aidi station is characterized by vertisol soil type with annual precipitation of about 300 mm over the two years. The Marchouch station is characterized by cambisol soil type with annual precipitation of about 400 mm over the two years. The stations are characterized by moderate humidity with an annual temperature ranging between 10 and 40 °C. The panel was seeded at a rate of 100 kg ha^−1^ during the first week of December.

### 4.2. Phenotyping and Statistical Analyses

The GWAS panel was phenotyped for 11 traits: grain yield (GY), days to heading (DHE), days to maturity (DMA), plant height (PLH), number of tillers per plant (NTP), number of plants in m^2^ (NPM), number of seeds per spike (NSS), biomass, chlorophyll content (CC), canopy temperature (CT), and thousand kernel weight (TKW). After threshing each plot, GY was measured in kg/plot and then converted to t/ha. When 50% and 90% of the plants reached the heading and maturity phases, respectively, DHE and DMA were recorded for each plot. When the plants reached maturity, the PLH in centimeters was recorded in each plot by measuring the distance from the soil surface to the top of the spike, without measuring the length of the awns. The biomass was measured for all plots in kilograms. The leaf chlorophyll content was measured on the flag leaf using a SPAD-502Plus chlorophyll meter at the heading stage on a clear day between 9:00 and 12:00. Canopy temperature was measured at the booting stage using a thermometer. TKW was calculated in grams using a grain counter. The “RcmdrMisc” package in R was used to determine the means, maximum, minimum, and standard deviation, as well as the coefficient of variance.

### 4.3. Genotyping

Fresh leaf samples were used to obtain genomic DNA from 2-week-old seedlings. Before extraction, the samples were frozen in glace liquid and maintained at −80 °C according to the methodology outlined in [48]. A total of 15,735 SNP markers were retained after quality filtering (the markers were used to generate the genotypic data for the panel used).

### 4.4. Linkage Disequilibrium Analyses and Population Structure

Linkage disequilibrium (LD) was calculated using Tassel software, where LD values of the 200 bread wheat genotypes were investigated using the SNP markers distributed across wheat genomes. Remington and his collaborator’s approach was used to conduct the LD decay study [49]. The population structure was estimated as a principal component (PC) analysis.

### 4.5. Genome-Wide Association Mapping and Genes Annotation

Genome-wide association analysis for grain yield and yield-related traits was carried out using Genomic Association and Prediction Integrated Tool version 3 (GAPIT 3) in the R environment [50]. A kinship K matrix and PCs (MLM: PCs+K) were used in the analysis. The important markers were shown on Manhattan plots, and markers with a −Log10(p) > 3.0 were designated as significant MTAs across the 21 chromosomes. Only significant markers (−log10p > 3.0) were traced using the CMplot program in the R environment [51]. Functional genes associated with grain yield and yield-related traits were identified using the Triticum aestivum genome annotation files downloaded from the Ensemble Plants database (http://plants.ensembl.org/Triticum aestivum/Info/Index , accessed on 5 April 2022 and the National Center for Biotechnology Information (NCBI) database. Uniport (https://www.uniprot.org, accessed on 6 April 2022) was used to identify protein function.

## 5. Conclusions

Some genotypes demonstrated good grain yield and acceptable agronomic performance in this study, and they were included in the worldwide nurseries given by ICARDA to national partners in the CWANA and SSA areas for release and future usage as crossing block parents. Using 200 spring bread wheat genotypes, we discovered 104 MTAs and 46 genes linked with various agronomic parameters under rain-fed and irrigated conditions. The markers discovered in this work will be utilized for marker-assisted selection after validation using a new set of elite genotypes.

## Figures and Tables

**Figure 1 plants-11-02705-f001:**
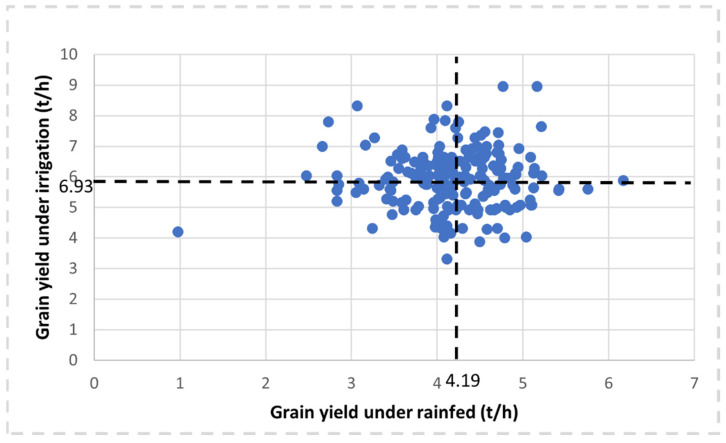
Scatter plots of grain yield under rainfed and irrigated conditions at Sidi El-Aidi station for 200 bread wheat genotypes grown during two seasons (2020 and 2021).

**Figure 2 plants-11-02705-f002:**
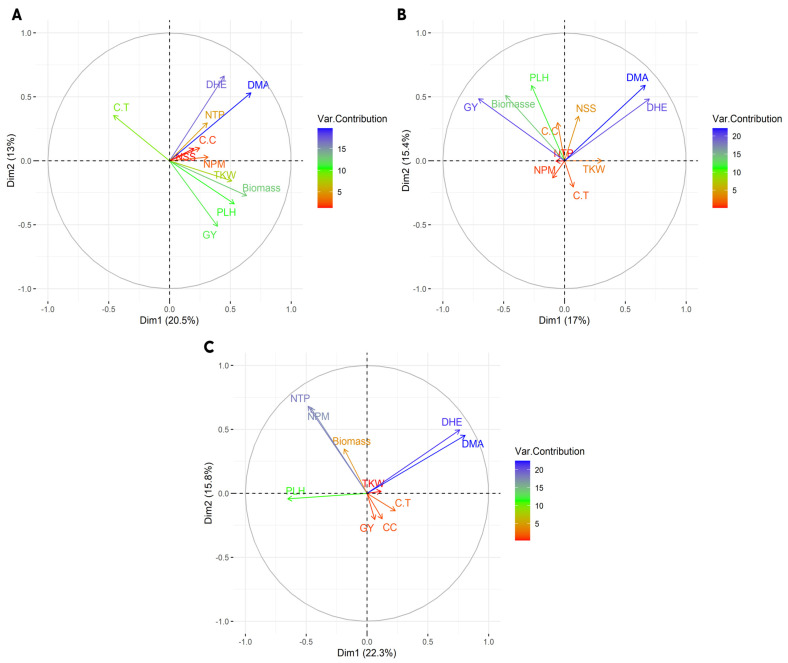
Principal component bi-plot analysis of 200 bread wheat genotypes grown during two seasons (2020 and 2021) under the rainfed regime (**A**) irrigated regime, (**B**) at Sidi Al-Aidi station and rainfed regime at Merchouch station, (**C**) in Morocco.

**Figure 3 plants-11-02705-f003:**
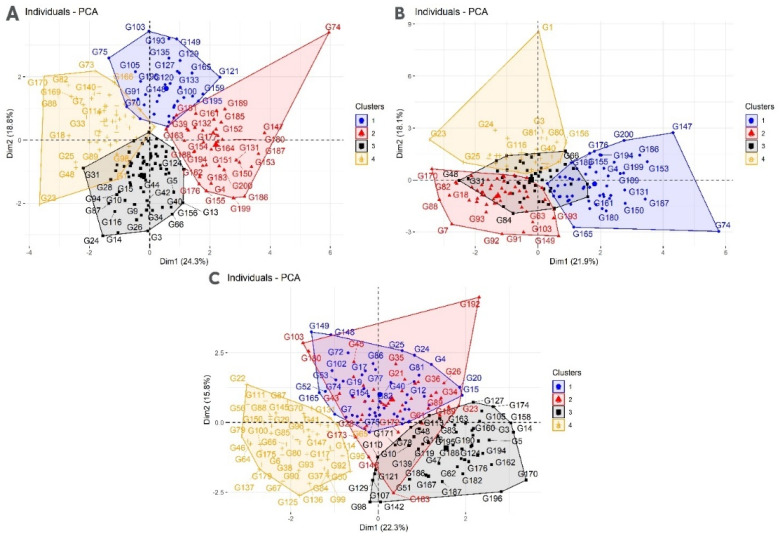
Cluster of 200 bread wheat genotypes grown during two seasons (2018 and 2019) under the rainfed regime (**A**) irrigated regime, (**B**) at Sidi Al-Aidi station, and rainfed regime at Merchouch station (**C**) in Morocco.

**Figure 4 plants-11-02705-f004:**
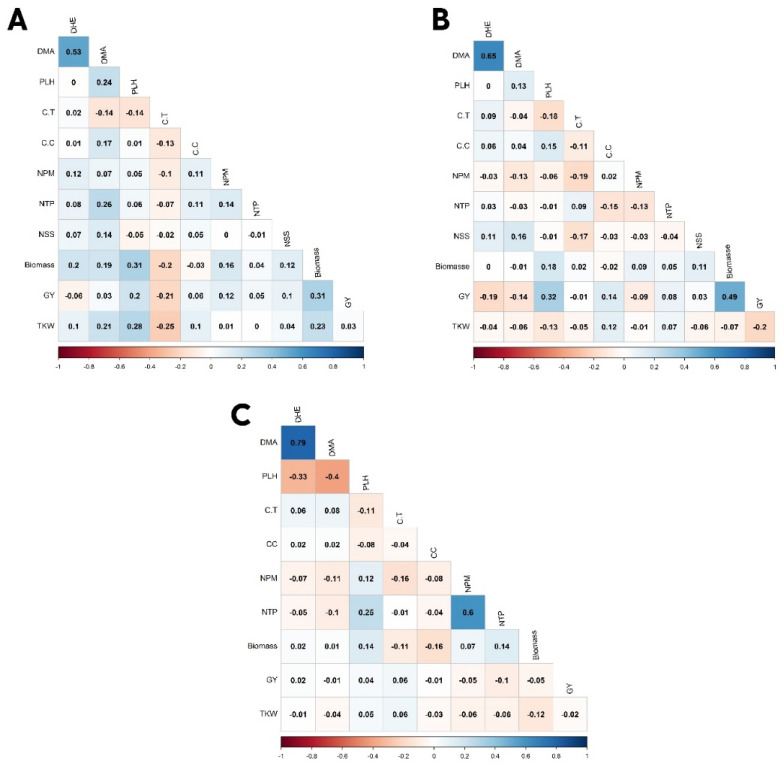
Correlation for agronomic and physiology traits under rainfed (**A**) and irrigated (**B**) regime at Sidi Al-Aidi station and rainfed regime at Merchouch station (**C**) in Morocco.

**Figure 5 plants-11-02705-f005:**
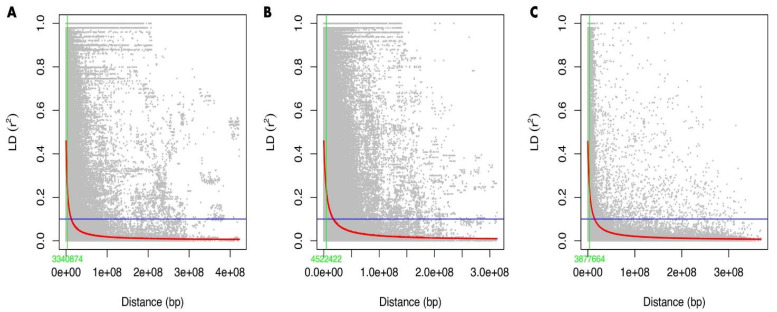
Linkage disequilibrium decay for A genome (**A**), B genome (**B**), and D genome (**C**) of 200 bread wheat genotypes.

**Figure 6 plants-11-02705-f006:**
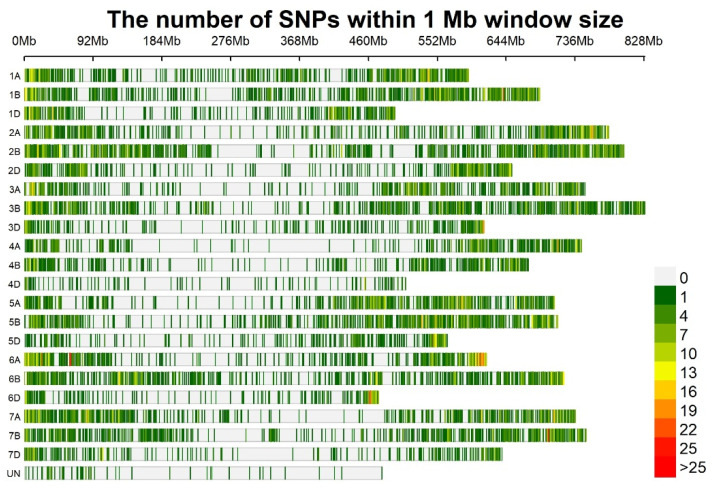
Marker density on each bread wheat chromosome of 200 bread wheat genotypes.

**Figure 7 plants-11-02705-f007:**
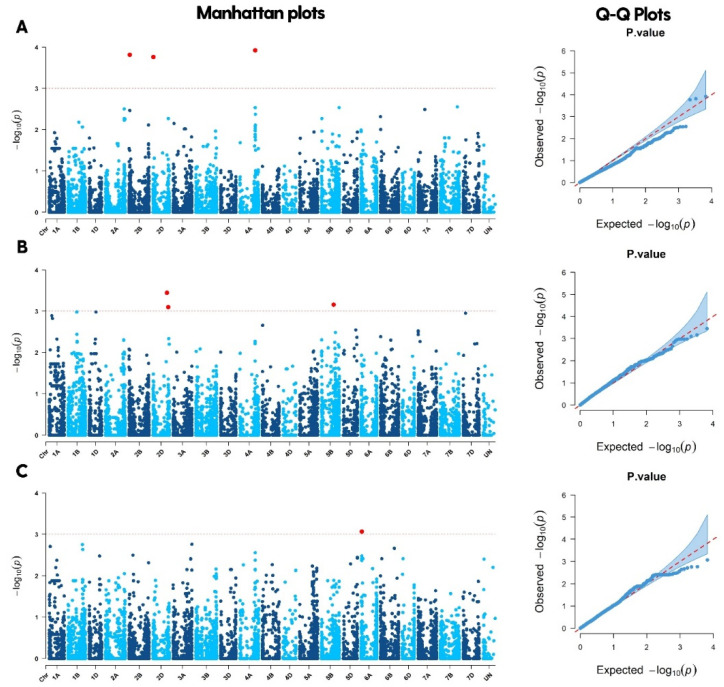
Manhattan plots and QQ plots of 200 bread wheat genotypes for grain yield under rainfed conditions at Sidi Al-Aidi station (**A**), irrigation conditions at Sidi Al-Aidi station (**B**), and rainfed conditions at Merchouch station (**C**).

**Figure 8 plants-11-02705-f008:**
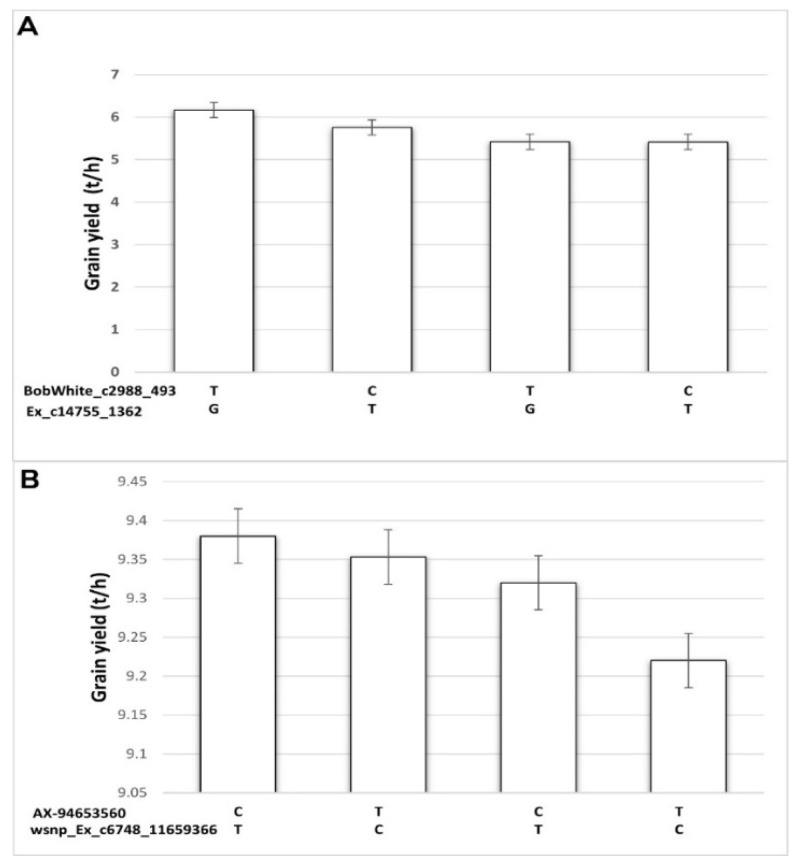
Allele combination of average grain yield from two field experiments performed in (**A**) rainfed conditions and (**B**) irrigated conditions at Sidi Al-Aidi station with 200 bread wheat genotypes having four base–pair combinations of two independent single nucleotide polymorphism markers associated with grain yield: BobWhite_c2988_493 and Ex_c14755_1362 in rainfed (**A**) and AX-94653560 and wsnp_Ex_c6748_11659366 in irrigated (**B**).

**Table 1 plants-11-02705-t001:** Phenotypic variation for grain yield and yield-related traits of the 200 bread wheat genotypes under rainfed and irrigated conditions.

Location	Condition	Trait	Mean	MIN	MAX	SD	CV	H^2^	<0.001
Sidi Al-Aidi	Irrigated	DHE	69.79	62	89	3.07	4.41	0.72	***
		DMA	114.55	108	136	2.76	2.41	0.65	***
		PLH (cm)	79.44	67.5	103.5	1.39	6.94	0.70	***
		C.T (°C)	23	21.3	26.4	0.72	3.13	0.52	***
		C.C (SPAD)	11.5	10.65	13.2	0.36	3.13	0.57	***
		NP/m^2^	32.62	20	60	8.46	2.59	0.53	***
		NTP	5.39	3	9	1.25	23.28	0.67	***
		NSS	59.21	31	97	12.19	20.59	0.8	***
		Biomass (kg)	2.97	1.38	5.4	0.85	22.68	0.77	***
		GY (t/h)	5.93	3.31	9.38	0.95	0.16	0.45	***
		TKW (g)	38.62	33	45.8	0.16	5.95	0.54	***
Sidi Al-Aidi	Rainfed	DHE	66.47	54	86	4.22	6.36	0.62	***
		DMA	108.65	106	123	4.08	3.76	0.52	***
		PLH (cm)	70.53	23	85.5	6.96	9.86	0.65	***
		C.T (°C)	23.96	22.27	32.67	0.85	3.57	0.42	***
		C.C (SPAD)	47.86	40.82	53.05	2.19	4.58	0.51	***
		NP/m^2^	38.86	30	50	4.3	11.08	0.49	***
		NTP	3.62	3	5	0.46	12.73	0.46	***
		NSS	43.85	22	76	10.02	22.85	0.63	***
		Biomass (kg)	1.1	0.61	1.88	0.22	20.39	0.61	***
		GY (t/h)	4.19	0.97	6.16	0.63	15.04	0.32	***
		TKW (g)	19.70	14.4	24.6	1.64	8.33	0.48	***
Merchouch	Rainfed	DHE	59.90	53	89	3.82	6.38	0.64	***
		DMA	119.38	111	126	3.54	2.97	0.53	***
		PLH (cm)	62.97	42	81	7.16	11.37	0.67	***
		C.T (°C)	25.67	23.1	29.7	0.94	3.69	0.45	***
		C.C (SPAD)	49.89	41.9	60	3.43	6.88	0.55	***
		NP/m^2^	42.27	24	54	6.08	14.39	0.53	***
		NTP	5.63	4	8	0.84	15	0.48	***
		Biomass (kg)	3.51	1.65	5.37	0.59	16.85	0.70	***
		GY (t/h)	4.09	2.32	6.85	0.74	18.29	0.42	***
		TKW (g)	34.31	27	43	2.95	8.59	0.51	***

SD, standard of deviation; CV, coefficient of variation; H^2^, heritability; GY, grain yield, DHE, days to heading, DMA, days to maturity; PLH, plant height; NTP, number of tillers per plant; NPM, number of plants in m^2^; NSS, number of seeds per spike; CC, chlorophyll content; CT, canopy temperature; and TKW, thousand kernel weight; *** Significant at *p* ≤ 0.001.

**Table 2 plants-11-02705-t002:** List of the significant markers associated with Biomass, CC, CT, DHE, DMA, GY, NPM, NSS, NTP, and PLH from genome-wide association analysis of 200 bread wheat genotypes under the rainfed conditions at Sidi Al-Aidi station.

Trait	SNP Marker	Chromosome	Position (bp)	MAF	Marker Effect	R^2^	−Log10(p)	Reference
Biomass	Ex_c14755_1362	2D	20,769,480	0.37	0.07	0.1	4.17	[10]
Biomass	BobWhite_c2988_493	2B	33,839,742	0.37	−0.07	0.1	4.17	
Biomass	AX-94513795	3A	568,142,443	0.03	−0.20	0.08	3.39	[11]
Biomass	AX-94866541	3A	568,383,306	0.03	−0.20	0.08	3.39	
Biomass	Kukri_rep_c104865_57	5D	562,121,917	0.03	−0.23	0.08	3.38	[10]
Biomass	wsnp_Ra_c19083_28215239	2B	33,939,928	0.32	−0.07	0.08	3.32	[10]
Biomass	AX-95171082	2D	28,417,610	0.02	−0.26	0.08	3.19	[12]
Biomass	RAC875_c41613_124	6D	163,836,060	0.02	−0.26	0.08	3.19	[13]
Biomass	Tdurum_contig51605_194	3B	822,682,075	0.04	−0.16	0.08	3.02	[14]
Biomass	BS00071948_51	3A	523,554,827	0.04	−0.16	0.08	3.02	[10]
Biomass	IACX6092	3A	519,207,590	0.04	−0.16	0.08	3.02	[10]
CC	IAAV108	5A	601,615,667	0.4	0.59	0.08	3.92	
CC	wsnp_Ku_c8400_14280021	3A	659,159,810	0.47	0.53	0.08	3.62	[15]
CC	Kukri_c51474_334	1B	457,378,726	0.24	−0.60	0.08	3.58	[16]
CT	AX-94463982	7A	53,911,180	0.04	−0.95	0.08	7.03	
CT	Tdurum_contig49841_618	5B	38,166,746	0.04	−0.82	0.08	5.97	[17]
CT	RAC875_c43581_280	6B	199,014,751	0.03	0.88	0.08	5.55	[13]
CT	Tdurum_contig30403_411	6D	256,247,450	0.03	0.88	0.08	5.55	
CT	AX-94741870	6B	146,646,555	0.04	0.81	0.08	5.5	
CT	Tdurum_contig9584_463	7A	53,916,532	0.05	−0.72	0.08	5.5	
CT	GENE-4021_496	6A	610,349,884	0.05	0.60	0.08	4.36	[14]
CT	Tdurum_contig29607_294	6A	609,379,892	0.05	0.58	0.08	4.12	[18]
CT	AX-94613704	1B	579,797,780	0.05	−0.60	0.08	3.77	
CT	BS00064077_51	3D	15,029,534	0.07	−0.43	0.08	3.33	[13]
CT	BS00074911_51	1B	626,226,213	0.08	−0.44	0.08	3.29	[10]
CT	BobWhite_rep_c64068_241	2B	775,336,369	0.33	−0.22	0.08	3.09	
DHE	BS00065105_51	2B	69,648,943	0.11	−1.57	0.08	3.78	[19]
DHE	RAC875_c8271_1352	1B	56,312,090	0.44	−1.23	0.08	3.68	[20]
DMA	AX-95103761	5B	544,374,823	0.05	2.50	0.08	3.99	
DMA	AX-94382081	7D	553,220,240	0.02	−4.26	0.08	3.95	
GY	BobWhite_c2988_493	2B	33,839,742	0.37	−0.16	0.08	3.81	[10]
GY	Ex_c14755_1362	2D	20,769,480	0.37	0.16	0.08	3.76	
GY	RAC875_c17918_321	4A	610,640,806	0.25	0.16	0.08	3.92	[10]
NPM	Ra_c29200_300	1A	546,044,432	0.27	1.37	0.08	3.63	
NPM	AX-94711993	4A	624,925,453	0.35	−1.21	0.08	3.28	[11]
NPM	AX-94690045	1D	464,930,094	0.27	−1.26	0.08	3.26	[11]
NPM	AX-94822081	3A	48,648,359	0.27	−1.26	0.08	3.26	[11]
NPM	Ra_c5683_1762	1A	551,461,645	0.27	−1.26	0.08	3.26	[10]
NPM	Tdurum_contig18174_283	1A	566,484,682	0.27	−1.26	0.08	3.26	[21]
NPM	Tdurum_contig56662_54	1A	557,932,838	0.27	−1.26	0.08	3.26	[22]
NPM	wsnp_Ex_c23598_32826926	1A	567,979,210	0.27	−1.26	0.08	3.26	[21]
NPM	wsnp_Ku_rep_c109724_94227136	1A	564,302,427	0.32	1.20	0.08	3.11	[10]
NPM	AX-95077803	1A	569,947,743	0.28	1.23	0.08	3.09	
NPM	Excalibur_c66_147	1A	572,350,787	0.26	−1.26	0.08	3.08	[10]
NPM	wsnp_CAP11_c146_160903	1A	572,350,833	0.26	−1.26	0.08	3.08	[23]
NPM	Tdurum_contig81011_244	1A	572,350,882	0.26	1.26	0.08	3.08	[22]
NPM	wsnp_Ra_c5433_9630495	3D	603,264,029	0.04	−2.85	0.08	3.04	[24]
NPM	AX-95162673	1A	545,528,557	0.28	1.20	0.08	3.04	
NSS	Excalibur_c5329_1335	5B	580,686,253	0.22	4.24	0.08	4.85	[25]
NSS	BS00099719_51	5B	580,103,265	0.22	4.24	0.08	4.8	[26]
NSS	wsnp_Ra_c39562_47242455	5B	580,103,342	0.22	4.12	0.08	4.67	[27]
NSS	Excalibur_c9391_1016	5B	580,085,161	0.21	−4.18	0.08	4.56	[28]
NSS	wsnp_Ex_c3834_6971712	5B	536,516,286	0.16	−4.28	0.08	3.76	
NSS	CAP11_c991_160	6B	577,184,321	0.05	7.41	0.08	3.72	[10]
NSS	AX-94814963	7D	171,001,100	0.12	−4.49	0.08	3.6	
NSS	TA001769-0538	1D	461,140,579	0.09	4.69	0.08	3.3	[29]
NSS	BS00066944_51	1B	587,048,099	0.36	2.75	0.08	3.08	
NSS	TA005251-0278	1A	532,251,634	0.37	2.71	0.08	3.05	
NSS	AX-95257656	1B	586,552,915	0.37	−2.71	0.08	3.05	
NTP	AX-95169625	3B	54,757,975	0.2	−0.14	0.08	3.06	
PLH	BS00037225_51	3B	590,294,618	0.02	13.79	0.08	7.1	
PLH	AX-94552125	1B	11,692,172	0.01	9.70	0.08	6.03	
PLH	BS00072156_51	5A	540,611,537	0.05	−4.58	0.08	3.92	
PLH	BS00089597_51	5D	552,040,060	0.06	3.80	0.08	3.18	
PLH	AX-94532002	5B	704,500,660	0.21	−2.36	0.08	3.1	
PLH	Ra_c22675_581	4A	611,201,088	0.21	−2.36	0.08	3.1	
PLH	wsnp_Ku_c9763_16287132	6A	581,748,362	0.15	−2.97	0.08	3.05	
TKW	AX-94864643	4D	504,562,200	0.02	−2.73	0.08	3.5	[11]
TKW	AX-94512414	4A	738,514,655	0.03	2.03	0.08	3.28	[11]
TKW	BS00044443_51	7B	498,149,722	0.02	−2.23	0.08	3.21	[10]
TKW	Excalibur_rep_c102136_270	7B	498,519,645	0.02	2.23	0.08	3.21	[10]
TKW	AX-94389822	2A	188,904,785	0.02	2.62	0.07	3.08	
TKW	Kukri_rep_c69627_954	6A	584,678,686	0.04	−1.66	0.07	3.05	[10]
TKW	AX-94826800	5B	35,976,675	0.02	2.14	0.07	3.01	

MAF, minor allele frequency; bp, base pair.

**Table 3 plants-11-02705-t003:** List of the significant markers associated with Biomass, CC, CT, DHE, DMA, GY, NPM, NSS, NTP, PLH, and TKW from genome-wide association analysis of 200 bread wheat genotypes under the irrigated conditions at Sidi Al-Aidi station.

Trait	SNP Marker	Chromosome	Position (bp)	MAF	Marker Effect	R^2^	−Log10(p)	Reference
Biomass	BS00022169_51	7A	691,259,651	0.02	−0.93	0.08	4.14	[30]
Biomass	wsnp_Ku_c4035_7363089	7A	688,686,324	0.02	−0.93	0.08	4.14	
Biomass	wsnp_Ra_c19741_28965647	7A	688,688,826	0.02	−0.93	0.08	4.14	
Biomass	AX-94870094	5B	208,440,868	0.03	−0.67	0.06	3.18	[11]
Biomass	wsnp_Ex_c6748_11659366	5B	484,735,572	0.03	−0.62	0.06	3.08	
CC	IAAV5819	3B	72,716,470	0.03	0.33	0.09	3.95	[10]
CC	AX-94826800	5B	35,976,675	0.02	−0.33	0.07	3.26	[11]
CC	BobWhite_c15352_394	7A	667,214,250	0.03	−0.31	0.07	3.25	
CC	RAC875_c41731_321	2A	470,238,226	0.39	−0.09	0.07	3.05	
CC	BS00003816_51	1D	435,801,502	0.18	0.11	0.07	3.05	[13]
CC	BS00040568_51	1D	435,928,205	0.18	−0.11	0.07	3.05	[13]
CC	Kukri_c35426_507	3A	639,086,463	0.31	−0.09	0.07	3.01	
CT	IAAV5819	3B	72,716,470	0.03	0.67	0.09	3.95	[10]
CT	AX-94826800	5B	35,976,675	0.02	−0.66	0.07	3.26	
CT	BobWhite_c15352_394	7A	667,214,250	0.03	−0.63	0.07	3.25	
CT	RAC875_c41731_321	2A	470,238,226	0.39	−0.19	0.07	3.05	
CT	BS00003816_51	1D	435,801,502	0.18	0.22	0.07	3.05	[13]
CT	BS00040568_51	1D	435,928,205	0.18	−0.22	0.07	3.05	[13]
CT	Kukri_c35426_507	3A	639,086,463	0.31	−0.19	0.07	3.01	[10]
DHE	Kukri_rep_c106285_295	1D	4,235,838	0.05	−2.06	0.09	3.81	[10]
DMA	BS00063748_51	2A	76,344,906	0.23	−0.83	0.07	3.57	[13]
DMA	BS00076261_51	2D	76,580,003	0.23	−0.83	0.07	3.77	[13]
DMA	wsnp_BE488779D_Ta_1_2	2D	76,639,717	0.23	−0.83	0.07	3.24	
GY	AX-94653560	2D	552,989,487	0.27	0.72	0.07	3.44	[11]
GY	wsnp_Ex_c6748_11659366	5B	484,735,572	0.03	−1.22	0.06	3.15	
GY	AX-94430599	2D	601,601,896	0.47	−0.42	0.06	3.10	[11]
NPM	Excalibur_rep_c69730_391	5B	566,504,645	0.11	4.76	0.12	4.87	
NPM	Kukri_c65380_490	2D	605,370,885	0.06	5.40	0.1	4.12	
NPM	Excalibur_c23239_783	2A	735,005,978	0.02	6.90	0.08	3.35	
NPM	Excalibur_c23239_961	2D	602,513,120	0.02	6.90	0.08	3.35	
NPM	Excalibur_c5442_1691	6A	16,571,210	0.09	−4.05	0.08	3.33	
NPM	AX-94637605	5A	502,799,409	0.08	−4.20	0.08	3.25	[11]
NSS	Kukri_c7786_81	5D	546,689,186	0.22	4.05	0.07	3.51	[31]
NSS	Kukri_c19883_629	4A	732,519,006	0.04	8.37	0.07	3.19	
NSS	BS00022036_51	5D	547,273,760	0.23	−3.71	0.07	3.18	[13]
NSS	Excalibur_c64287_145	4A	622,200,799	0.23	−3.71	0.07	3.18	
NSS	CAP12_c5949_104	5D	546,650,824	0.23	3.68	0.06	3.05	[31]
NTP	IAAV5776	1B	675,560,923	0.11	0.45	0.06	3.26	[10]
NTP	BS00063512_51	1B	676,192,103	0.09	−0.48	0.06	3.69	[13]
PLH	JD_c1314_1184	7A	706,832,129	0.08	−2.37	0.08	3.07	
TKW	BS00064829_51	1B	12,813,223	0.17	0.67	0.07	4.17	[13]
TKW	Kukri_c2446_683	1D	7,353,184	0.10	−0.76	0.07	3.82	
TKW	Kukri_rep_c69910_1153	1A	11,937,614	0.17	0.61	0.07	3.75	
TKW	BS00069300_51	1A	12,006,112	0.18	0.61	0.07	3.73	[13]
TKW	Excalibur_c44883_244	1A	11,940,136	0.17	0.58	0.07	3.41	
TKW	BS00003761_51	1A	16,390,705	0.17	0.57	0.07	3.30	[13]
TKW	AX-94742021	1A	14,361,026	0.17	0.57	0.07	3.11	[11]
TKW	Excalibur_c25891_1402	3B	554,033,860	0.21	0.54	0.07	3.09	

MAF, minor allele frequency; bp, base pair.

**Table 4 plants-11-02705-t004:** List of the significant markers associated with Biomass, CC, CT, DHE, DMA, GY, NPM, and TKW from genome-wide association analysis of 200 bread wheat genotypes under rainfed conditions at Merchouch station.

Trait	SNP Marker	Chromosome	Position (bp)	MAF	Marker Effect	R^2^	−Log10(p)	Reference
Biomass	BS00099879_51	6A	608,074,450	0.09	−0.34	0.01	4.10	[13]
Biomass	AX-95226309	6D	460,572,468	0.09	−0.30	0.01	3.88	[11]
Biomass	Kukri_c42895_593	6B	65,732,454	0.36	0.17	0.01	3.31	
CC	AX-95126745	4A	5,464,991	0.49	−1.08	0.07	4.48	[11]
CC	BobWhite_rep_c64913_315	5A	413,418,596	0.38	1.05	0.07	3.38	
CC	Excalibur_rep_c108030_260	4D	108,902,883	0.41	−0.94	0.07	3.30	[13]
CC	Excalibur_c2171_2728	5A	708,441,404	0.27	−1.15	0.07	3.24	
CC	AX-94974108	5A	708,309,244	0.26	1.15	0.07	3.23	[11]
CC	wsnp_Ex_c2171_4074003	5A	708,442,382	0.26	−1.15	0.07	3.23	
CC	Excalibur_rep_c106790_155	4D	113,155,346	0.4	−0.93	0.07	3.18	[13]
CC	AX-94483885	4D	98,582,176	0.39	−0.92	0.07	3.16	[11]
CC	AX-95231592	5A	708,165,083	0.25	1.13	0.07	3.12	[11]
CC	Excalibur_c42255_425	5A	702,166,657	0.31	1.04	0.07	3.09	[13]
CC	Kukri_c16087_281	5A	702,166,100	0.31	1.04	0.07	3.09	
CC	wsnp_Ex_c3764_6853627	2B	185,761,088	0.49	0.95	0.07	3.00	
CT	AX-94419426	1B	677,254,971	0.01	−1.40	0.003	4.34	[11]
CT	AX-94796020	1A	491,135,196	0.01	−1.40	0.003	4.34	[11]
CT	AX-94975215	3D	613,735,962	0.01	−1.40	0.003	4.34	[11]
CT	AX-95152594	3B	18,518,629	0.03	−0.78	0.003	3.45	[11]
CT	wsnp_JD_c20555_18262260	7A	674,276,748	0.07	−0.49	0.003	3.41	[32]
CT	RAC875_c8565_926	7A	725,929,902	0.04	0.65	0.003	3.4	
CT	AX-94513795	3A	568,142,443	0.02	0.87	0.003	3.4	[11]
CT	AX-94866541	3A	568,383,306	0.02	0.87	0.003	3.4	[11]
CT	AX-94630410	6D	249,961,124	0.02	−0.97	0.003	3.2	[11]
CT	AX-95023231	2D	640,215,412	0.02	−0.82	0.003	3.07	[11]
CT	BS00043716_51	6A	230,706,086	0.02	−0.81	0.003	3.04	
DHE	IAAV2346	5B	17,968,969	0.04	−2.89	0.04	3.63	[10]
DHE	AX-95168017	3B	648,927,053	0.03	3.53	0.04	3.49	[11]
DHE	BS00068817_51	3B	607,812,618	0.03	−3.53	0.04	3.49	[13]
DHE	BS00068816_51	3B	607,812,579	0.02	3.42	0.04	3.43	[13]
DHE	Excalibur_c48047_90	3B	617,682,876	0.02	−3.42	0.04	3.43	
DHE	BS00067651_51	5D	521,522,831	0.05	2.51	0.04	3.35	[13]
DHE	Kukri_rep_c108378_52	5B	583,178,232	0.06	−2.29	0.04	3.32	
DHE	AX-94792657	5A	596,013,518	0.06	2.29	0.04	3.32	
DHE	Excalibur_rep_c113405_180	3B	618,151,165	0.03	2.80	0.04	3.32	
DHE	GENE-3207_610	5B	17,545,461	0.04	−2.46	0.04	3.11	[33]
DHE	BS00065543_51	5B	17,575,009	0.04	2.33	0.04	3.07	
DHE	Excalibur_c11605_156	5B	17,945,028	0.04	2.33	0.04	3.07	[13]
DHE	GENE-3207_134	5B	17,545,728	0.04	−2.33	0.04	3.07	[33]
DHE	BS00039935_51	4B	5,468,520	0.03	−2.89	0.04	3.01	
DMA	Kukri_c8594_203	4B	17,255,199	0.09	−1.65	0.06	3.64	
DMA	tplb0035d20_506	4D	9,249,227	0.09	−1.65	0.06	3.64	
DMA	Excalibur_c45297_316	5A	571,784,879	0.24	1.19	0.06	3.45	[13]
DMA	wsnp_Ex_rep_c69647_68598463	5A	571,786,104	0.24	−1.19	0.06	3.45	
DMA	AX-94820753	5B	689,950,369	0.06	−1.78	0.06	3.08	[11]
DMA	wsnp_RFL_Contig2914_2757372	2B	740,805,174	0.38	1.04	0.06	3.01	
GY	Kukri_c264_438	6A	29,558,020	0.03	0.57	0.01	3.06	
NPM	IACX11794	7D	12,470,234	0.16	2.54	0.03	4.04	[34]
NPM	Excalibur_c7255_697	7D	13,602,327	0.11	−2.79	0.03	3.61	[13]
NPM	D_GB5Y7FA02IDDA9_183	7D	13,487,464	0.11	−2.73	0.03	3.55	[20]
NPM	BS00022449_51	UN	89,193,164	0.16	−2.35	0.03	3.53	[13]
NPM	Excalibur_c833_1405	3D	585,088,528	0.11	2.65	0.03	3.4	
NPM	RAC875_rep_c74271_414	5B	696,492,752	0.28	−1.88	0.03	3.04	
TKW	AX-94712739	2B	766,563,614	0.01	−3.37	0.005	3.94	[11]
TKW	BobWhite_c14486_122	5A	615,864,204	0.01	−3.37	0.005	3.8	

MAF, minor allele frequency; bp, base pair.

## Data Availability

Not applicable.

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
