# Peer review of "Genetic Dissection of Drought Tolerance of Elite Bread Wheat (*Triticum aestivum* L.) Genotypes Using Genome Wide Association Study in Morocco"

_plants, 2022, doi:10.3390/plants11202705_

Round 1

Reviewer 1 Report

The paper is devoted to important subject of genetic control of agronomic traits in wheat under variable moisture conditions. The study methodology and data obtained is very relevant. The study provides important contribution to science and need to be published. However, the authors invested substantial efforts in phenotyping 200 lines at 6 environments, however  the presentation of the needs substantial improvement following few points below.

11.      Line 12-13.  Unclear statement: “…challenge of terminal moisture and increase…”, probably moisture stress.

22.      Wheat production figures for Morocco would be useful.

33.      The phenotyping conditions shall mention presence of diseases and if they affected the germplasm performance.

44.      In section results the traits abbreviations are not explained at the first mention.  

55.      There are too many traits abbreviation. Commonly used abbreviations are fine like TKW or abbreviations of traits with three words. Many others can be avoided.

66.      Table 1 does not have heading.

77.      Section 2.1 can be presented in a more logical way by explaining agronomically the differences between sites, rainfed and irrigated trials and seasons.

88.      Section 2.2. Lines 117-123 are not necessary as the reader can see them in the graphs. The major issue here is analysis using the average values for two years. This is acceptable only if authors explain that the two seasons were very similar. By the way the G x E ANOVA is not presented and it shall indicate the effect of years and irrigation factor. This is an important section to emphasize from agronomy and breeding perspective the effect of moisture stress on traits relationship.

99.      Section 2.3. The authors do not mention if the analysis was conducted using average values for two seasons. This is hardly acceptable as analysis for each season and environment separate would provide more valuable information. Tables 2, 3 and 4 will benefit if the markers effects are presented even at expense of other parameters.

110.   Lines 228-243. This is very important discussion. The authors state: “…Under rainfed conditions at Sidi Al-Aidi station, out of the 3 common significant 228 MTAs found across 2 years..”. This means that GWAS was done for each season separate. Then the results in  the tables 2-4  are from which season? Are there markers with similar effects under rainfed and irrigated conditions?

111.   Section 3.1. Two similar statements: “Grain yield is a complicated property governed by numerous genes, the expression of which is controlled by environmental conditions.” “Grain yield is a complicated trait governed by several genes, the expression of which is affected 281 by environmental conditions.”

112.   Section 3.2. Reference to the same markers found by other studies is very important a may be added into tables 2-4.

113.   Conclusions. “Some genotypes demonstrated good grain yield and acceptable agronomic performance in this study…”. This is not sufficient. There is good justification to add a table with the best lines across different environments, include data for key traits and presence/absence on the key makers. Full list of material with pedigree and probably reference to international nursery and entry number can be presented as supplement.

114.   The authors shall defend reference to “drought” in the paper title. The rainfed trials with average yield of 3.5 and 4.2 t/ha are hardly drought environments. However, adaptation to variable moisture availability is also important. Morocco yield was 2.5 t/ha in 2018 and only 0.9 t/ha in 2020 – FAO.

Author Response

Thank you very much for reviewing our paper.

  1. Line 12-13.  Unclear statement: “…challenge of terminal moisture and increase…”, probably moisture stress.

Response 1: Challenge of terminal moisture stress and increase increase wheat productivity.

  1. Wheat production figures for Morocco would be useful.

Response 2: Done.

  1. The phenotyping conditions shall mention presence of diseases and if they affected the germplasm performance.

Response 3: We did not recorded any diseases during the agricultural season.

  1. In section results the traits abbreviations are not explained at the first mention.

Response 4: Done.

  1. There are too many traits abbreviation. Commonly used abbreviations are fine like TKW or abbreviations of traits with three words. Many others can be avoided.

Response 5: Done.

  1. Table 1 does not have heading.

Response 6: Done.

  1. Section 2.1 can be presented in a more logical way by explaining agronomically the differences between sites, rainfed and irrigated trials and seasons.

Response 7: Done.

  1. Section 2.2. Lines 117-123 are not necessary as the reader can see them in the graphs. The major issue here is analysis using the average values for two years. This is acceptable only if authors explain that the two seasons were very similar. By the way the G x E ANOVA is not presented and it shall indicate the effect of years and irrigation factor. This is an important section to emphasize from agronomy and breeding perspective the effect of moisture stress on traits relationship.

Response 8: The two agricultural seasons were approximately similar, because of that we did this analysis.

  1. Section 2.3. The authors do not mention if the analysis was conducted using average values for two seasons. This is hardly acceptable as analysis for each season and environment separate would provide more valuable information. Tables 2, 3 and 4 will benefit if the markers effects are presented even at expense of other parameters.

Response 9: we mentioned the results of GAWS is the mean of two years (line 169).

we putted the marker effect on the tables 2,3 and 4.

  1. Lines 228-243. This is very important discussion. The authors state: “…Under rainfed conditions at Sidi Al-Aidi station, out of the 3 common significant 228 MTAs found across 2 years.”. This means that GWAS was done for each season separate. Then the results in the tables 2-4 are from which season? Are there markers with similar effects under rainfed and irrigated conditions?

Response 10: We mean the highest significant markers for the GWAS of the mean of two years, I corrected it. The GWAS was done of the mean of two years.

There are 6 markers with similar effects, where 2x2x2 markers in the same environment. We mentioned them in the manuscript.

  1. Section 3.1. Two similar statements: “Grain yield is a complicated property governed by numerous genes, the expression of which is controlled by environmental conditions.” “Grain yield is a complicated trait governed by several genes, the expression of which is affected 281 by environmental conditions.”

Response 11: We deleted the first sentence and we keep the second one.

  1. Section 3.2. Reference to the same markers found by other studies is very important a may be added into tables 2-4.

Response 12: Done.

  1. Conclusions. “Some genotypes demonstrated good grain yield and acceptable agronomic performance in this study…”. This is not sufficient. There is good justification to add a table with the best lines across different environments, include data for key traits and presence/absence on the key makers. Full list of material with pedigree and probably reference to international nursery and entry number can be presented as supplement.

Response 13: We putted the best genotypes with their pedigree for the audience in the supplementary material. Also, I putted all genotypes in the supplementary material.

  1. The authors shall defend reference to “drought” in the paper title. The rainfed trials with average yield of 3.5 and 4.2 t/ha are hardly drought environments. However, adaptation to variable moisture availability is also important. Morocco yield was 2.5 t/ha in 2018 and only 0.9 t/ha in 2020 – FAO.

Response 14: We mentioned in the title that the experiment was carried out under drought condition. Yes, you are right the rainfed trials with average yield of 3.5 and 4.2 t/ha are hardly drought environments. These last years, Morocco record low precipitation which led to low wheat production and also Morocco use cultivars that have  moderate grain yield. In ICARDA we developed so many genotypes that are drought tolerant with acceptable grain yield. 

Reviewer 2 Report

1. The abbreviation MLM corresponds to Mixed Lienar Model, and  you use Linear Mixed Model. 

2. Figure 2 comments in (2.2. Principal Components Analysis (PCA) and Correlation) do not match the values in the figure.

3.  In (2.3. Linkage Disequilibrium and Marker Trait Associations) there are many inconsistencies in the comments with the data in the tables. The number of MTАs does not match that in the tables. In the Merchouch station comment, the table should be Table 4.

4. In the same point 2.3 in the sentence "NTP trait recorded one significant marker located on chromosome 3B." , 3B should be 5B. In the next sentence тhe type of trait is not specified.

5.In the sentence "DHE recorded one significant marker located on chromosome 1D with −Log10(p) = 3.081.", the value for −Log10(p) is different from that in the table.

6. Тhe number of MTAs for CC and DMA in this point for rainfed conditions at Merchouch station is different from that in the table.

7. On Figure 7 (B) the markers are located on 2D, but you indicated 2D and 2B.

8. Check the text for grammatical errors. There are some sentences that start with a lowercase letter.

Author Response

Thank you very much for reviewing our paper.

  1. The abbreviation MLM corresponds to Mixed Lienar Model, and  you use Linear Mixed Model. 

Response 1 : Has been corrected.

  1. Figure 2 comments in (2.2. Principal Components Analysis (PCA) and Correlation) do not match the values in the figure.

Response 2 : Has been corrected.

  1. In (2.3. Linkage Disequilibrium and Marker Trait Associations) there are many inconsistencies in the comments with the data in the tables. The number of MTАs does not match that in the tables. In the Merchouch station comment, the table should be Table 4.

Response 3 : Has been corrected.

  1. In the same point 2.3 in the sentence "NTP trait recorded one significant marker located on chromosome 3B." , 3B should be 5B. In the next sentence тhe type of trait is not specified.

Response 4 : After verification the chromosome is 3B. The next sentence the trait is plant height.

5.In the sentence "DHE recorded one significant marker located on chromosome 1D with −Log10(p) = 3.081.", the value for −Log10(p) is different from that in the table.

Response 5 : Has been corrected.

  1. Тhe number of MTAs for CC and DMA in this point for rainfed conditions at Merchouch station is different from that in the table.

Response 6 : Has been corrected.

  1. On Figure 7 (B) the markers are located on 2D, but you indicated 2D and 2B.

Response 7 : Rainfed Sidi Alaidi station recorded 3 MTAs located on chromosomes 2B, 2D and 4A. while irrigated Sidi Alaidi station recorded 3 MTAs located on chromosomes 2D, 2D and 2A.

  1. Check the text for grammatical errors. There are some sentences that start with a lowercase letter.

Response 8 : Done.
